# Community engagement for birth preparedness and complication readiness in the Community Level Interventions for Pre-eclampsia (CLIP) Trial in India: a mixed-method evaluation

Avinash Kavi ![ORCID],[1] Mai-Lei Woo Kinshella ![ORCID],[2] Umesh Y Ramadurg,[3] Umesh Charantimath ![ORCID],[1] Geetanjali M Katageri,[3] Chandrashekhar C Karadiguddi,[1] Narayan V Honnungar,[1] Shashidhar G Bannale,[3] Geetanjali I Mungarwadi,[1] Jeffrey N Bone ![ORCID],[4] Marianne Vidler ![ORCID],[4] Laura Magee,[5] Ashalata Mallapur,[3] Shivaprasad S Goudar ![ORCID],[1] Mrutyunjaya Bellad ![ORCID],[1] Richard Derman,[6] Peter von Dadelszen ![ORCID],[7] The CLIP India Working Group[1,4,7]

For numbered affiliations see end of article.

**Correspondence to**
Dr Avinash Kavi;
dravinashkavi@gmail.com

## ABSTRACT

**Objective**  To describe the process of community engagement (CE) in northern Karnataka, India and its impact on pre-eclampsia knowledge, birth preparedness and complication readiness, pregnancy-related care seeking and maternal morbidity.

**Design**  This study was a secondary analysis of a cluster randomised trial of Community Level Interventions for Pre-eclampsia (CLIP). A total of 12 clusters based on primary health centre catchment areas were randomised to intervention or control. CE was conducted in intervention clusters. CE attendance was summarised according to participant group using both quantitative and qualitative assessment. Pre-eclampsia knowledge, birth preparedness, health services engagement and perinatal outcomes was evaluated within trial surveillance. Outcomes were compared between trial arms using a mixed effects logistic regression model on RStudio (RStudio, Boston, USA). Community feedback notes were thematically analysed on NVivo V.12 (QSR International, Melbourne, Australia).

**Setting**  Belagavi and Bagalkote districts in rural Karnataka, India.

**Participants**  Pregnant women and women of reproductive age, mothers and mothers-in-law, community stakeholders and male household decision-makers and health workers.

**Results**  A total of 1379 CE meetings were conducted with 39 362 participants between November 2014 and October 2016. CE activities may have had an effect on modifying community attitudes towards hypertension in pregnancy and its complications. However, rates of pre-eclampsia knowledge, birth preparedness, health services engagement and maternal morbidities among individual pregnant women were not significantly impacted by CE activities in their area.

**Conclusion**  Evaluation of our CE programme in India demonstrates the feasibility of reaching pregnant women alongside household decision-makers, community stakeholders and health workers. More research is needed to

## STRENGTHS AND LIMITATIONS OF THIS STUDY

⇒ We systematically evaluated the effectiveness of a large-scale community engagement programme conducted with the Community Level Interventions for Pre-eclampsia (CLIP) Trial in rural India using a process evaluation framework. The logic model outlines the hypothesised relationships between inputs, outputs and outcomes of interest to explore potential pathways of change.

⇒ Data for the assessment were obtained from a prospective population-based surveillance system of women in the CLIP Trial and community engagement records including community feedback notes. Our study triangulated qualitative and quantitative datasets in a convergent mixed-methods approach.

⇒ The community engagement programme and its evaluation involved a wide range of health workers, pregnant women, household decision-makers and community leaders. Group sessions with many stakeholders allowed for a diversity of perspectives. Unfortunately, we do not have data on specific individuals.

⇒ The limited timeframe within a clinical trial may have been too short to appropriately evaluate community-level behavioural change and more research is needed to better understand the influence of external factors on maternal healthcare seeking.

⇒ Community engagement was one of the components of a complex intervention evaluated in the CLIP Trial and interventions involving multiple factors make the impact assessment more complicated.

explore the pathways of impact between broad community mobilisation to strengthen support for maternal care seeking and clinical outcomes of individual pregnant women.

**Trial registration number**  NCT01911494.

## INTRODUCTION

Over 94% of maternal deaths occur in low-income and middle-income countries, primarily in South Asia and sub-Saharan Africa, and are preventable.[1 2] Maternal deaths relate primarily to delays in triage (ability of care provider and women to identify who is severely ill and requires urgent care), transport (ability to get women to appropriate care when needed) and treatment (ability to provide appropriate treatment when care accessed).[3] Increased birth preparedness and complication readiness (BPCR) can help mitigate against delays and ensure timely identification of the need for seeking skilled care and arrival at the appropriate facility for pregnancy complications. India has an estimated maternal mortality ratio (MMR) of 113 per 100 000 live births in 2016–2018, with pregnancy hypertension accounting for 7% of maternal deaths.[4] While India has achieved a 6.3% average annual rate of decline from 1997 to 2018, progress has varied by region.[4] Though the states in southern India are often categorised as high-performing with better maternal and perinatal health indicators than much of the country, the MMR in Karnataka is double the rate in neighbouring Kerala (92 vs 42 per 100 000 live births).[4]

BPCR interventions are strongly recommended by the WHO to increase skilled birth attendance and timely use of facility care for obstetric and neonatal complications.[5 6] According to the WHO, BPCR involves counselling pregnant women and their families on knowledge of pregnancy danger signs and developing a BPCR plan that includes identification of health facility and transport for birth and in case of complications, identification of a birth companion and preferred birth attendant, supplies to bring to the facility, funds for any expenses and identification of a support person to look after other children at home as well as compatible blood donors if needed.[5–7] Involvement of community members is critical in the implementation of community health programmes.[8] In India, raising community awareness to support BPCR complements the national programme of community health workers called Accredited Social Health Activists (ASHAs), which aims to connect women and children to primary care.[9] ASHAs identify pregnant women in their communities and accompany them to nearby primary health centre for registration and antenatal care (ANC) visits. However, previous research in Karnataka revealed that pregnant women rarely made healthcare decisions on their own, and other family members and community leaders were highly influential in decision-making.[10] Bringing community members together to raise awareness of maternal health issues mobilises wider support for maternal healthcare seeking and ASHAs' work in local communities.

Over the past few decades, community engagement (CE) has emerged as an increasingly effective strategy for harnessing community potential for health improvement.[11–16] CE is defined as 'a process of working collaboratively with groups of people who are affiliated by geographic proximity, special interests or similar situations, with respect to issues affecting their well-being'.[17] It is a dynamic relational process that facilitates communication, interaction, involvement and exchange between an organisation and a community for a range of social and organisational outcomes.[18] This strategy has been used for health promotion, research and policy making to address a variety of health issues.[19–22] CE enables a more contextualised understanding of community members' perceptions, thereby facilitating stronger relationships. Implementation of CE enhances the delivery of the healthcare services in an optimal manner for mothers and newborns, which ultimately aligns with the vision of national goal.

This paper is the secondary analysis of the Community Level Interventions for Pre-eclampsia (CLIP) Trial, India; the primary paper has been published elsewhere.[23] The paper describes the process of CE, one of the component of CLIP Trial, and evaluates the impact of CE on knowledge around pre-eclampsia, BPCR, pregnancy-related care seeking and rates of maternal morbidity.

## METHODS

### The CLIP Trial and study setting

The CLIP India Trial (Clinical Trials.gov NCT01911494) was a community-based cluster randomised controlled trial reported according to the Consolidated Standards of Reporting Trials 2010 Statement (online supplemental table S1, checklist). The CLIP India Trial was conducted in rural Karnataka, India where the incidence of pregnancy hypertension is 10.3%.[24] This study was conducted in 12 distinct geographic regions in Karnataka State, six in each Belagavi and Bagalkote districts (figure 1, map; figure 2, map). Belagavi has a population of 4 779 661 living in 1278 villages and Bagalkote has a population of 1 889 752 living in 627 villages.[25] There are an estimated 500 births annually in each district.[25] The adult female literacy rate varies in the two districts, at 70.2% in Belagavi, and 59.3% in Bagalkote, respectively.[26]

The primary health centre (PHC) and its catchment area served as the unit of randomisation. Clusters were chosen according to the region, accessibility for surveillance and absence of conflicting research activity. Clusters were allocated using a restricted randomisation algorithm, balancing region and population size to ensure equivalence between intervention and control clusters. Stratified randomisation was undertaken to allocate six clusters each to the intervention and control groups, using population size as a single stratification factor. Both intervention and control clusters were similar in terms of population (11.6 per hectare in intervention clusters vs 13.3 per hectare in control clusters) and community health workers (17 vs 18 ASHAs per cluster) density.[23] Among the pregnant women enrolled in the CLIP Trial (7839 in intervention clusters, 6944 in the control clusters), maternal age was on average 22–23 years old, 56%–58% of women had eight or more years

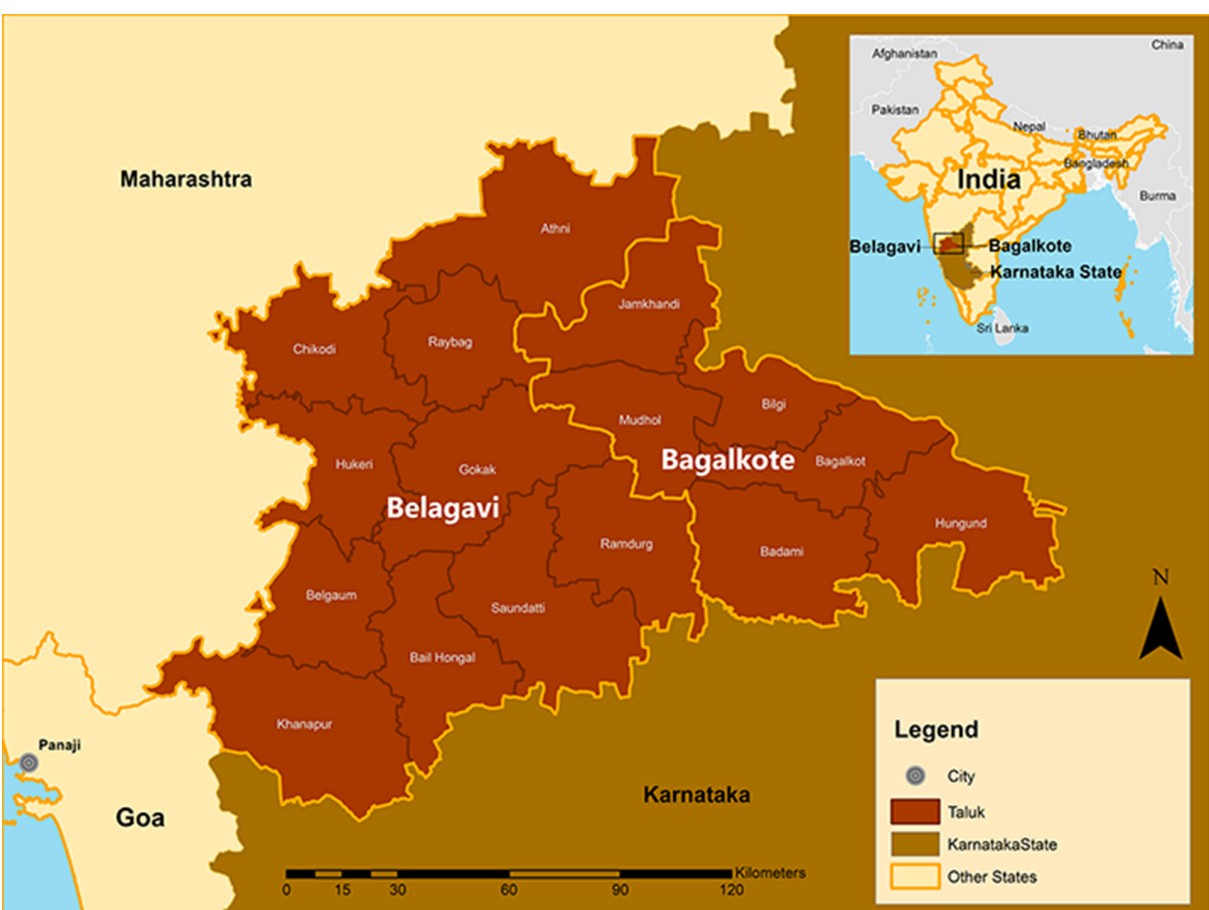

**Figure 1** Study location in India, Karnataka state.

of schooling and populations were over 90% Hindu without significant difference between intervention and control clusters.[23]

The overall CLIP Trial focused on implementing community-level evidence-based care to reduce all-cause maternal and perinatal mortality and major morbidity by supporting early identification and prompt referrals for pregnancy complications. The intervention included CE alongside strengthening clinical capacities of ASHAs to assess blood pressure and usage of PIERS-On-the-Move (POM). POM is a mobile-based application decision aid for community health workers to identify women at risk of adverse outcomes from pregnancy hypertension and guide management.[27 28] More details on the overall CLIP India Trial, participant characteristics and reporting on the primary outcomes have been published elsewhere.[23]

### Process of facilitating CE in the CLIP Trial

The CE component was carried out in intervention clusters over the entire trial period (November 2014 to October 2016). The CE activities aimed at creating culturally and contextually appropriate discussion to improve maternal health awareness and action around the prevention of maternal morbidity and mortality. CE aimed to improve birth preparedness, nutrition and promote appropriate ANC and

healthcare seeking. Topics included warning signs of pregnancy complications, permission to seek care, identifying a skilled birth attendant, facility for delivery and mode of transport as well as saving up funds for transport and treatment. Discussions also included adverse pregnancy outcomes and success stories in the community, overcoming local barriers to accessing maternal healthcare and the interventions in the CLIP Trial.

CE meetings were primarily held at the anganwadi (AW) centres in the village, which offer basic health services. These were locations where pregnant women and children routinely access ANC, immunisations and other maternal and child healthcare services. Meetings were also held at subcentres, PHCs, temples, public halls, schools or other appropriate nearby locations for large monthly sessions.

The activities convened multiple levels of the local healthcare system. This included workers Anganwadi Worker (AWW) at the AW centres, ASHAs who worked in the villages, auxiliary nurse midwives (ANMs) who worked at subcentres, lady health visitors (LHVs) who supervised the ANMs and medical officers who coordinated the PHC. In addition, ASHA supervisors and registry administrators (RAs) appointed as a part of the trial also attended meetings.

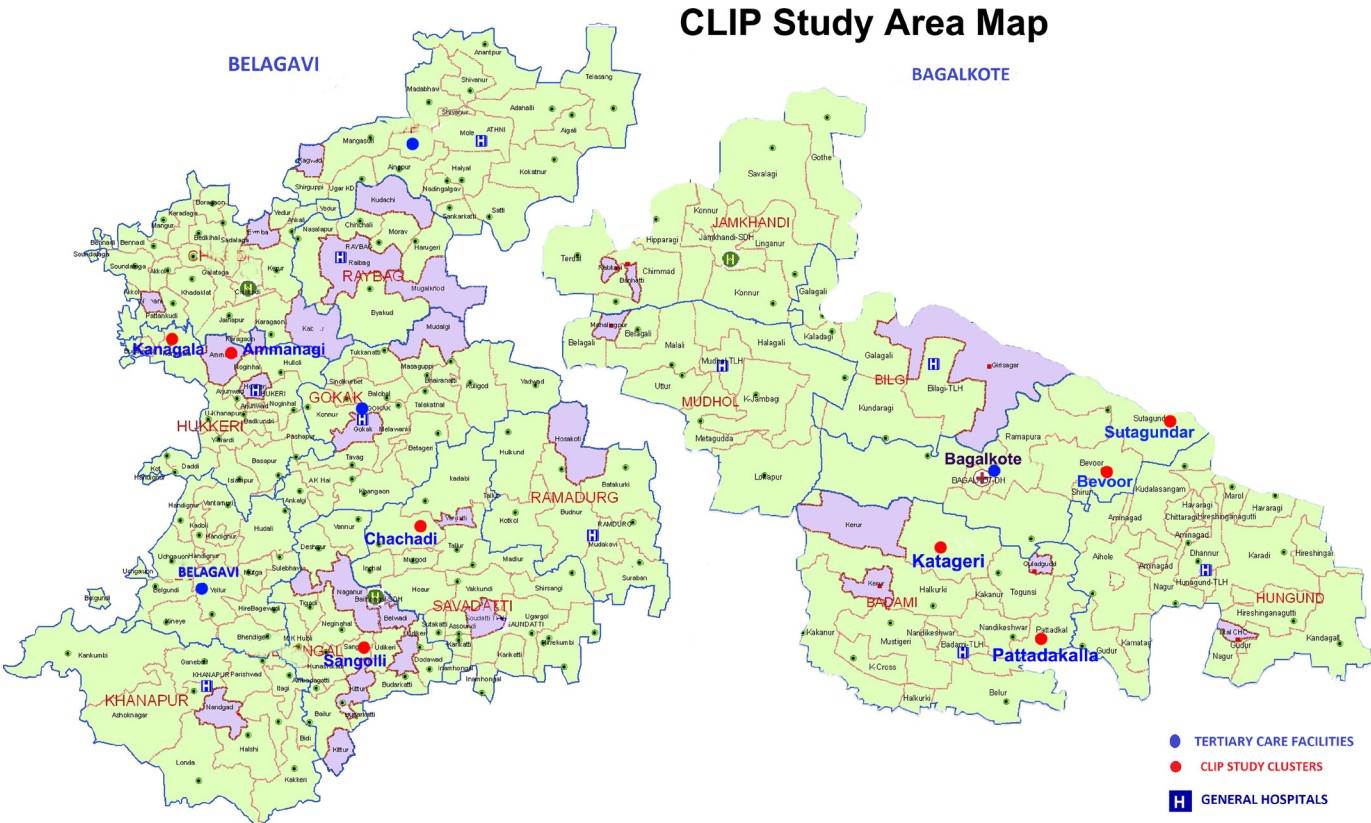

**Figure 2** Belagavi and Bagalkote districts, Karnataka state, India with detailed primary health centre areas. CLIP, Community Level Interventions for Pre-eclampsia.

Pregnant women and married women of reproductive age groups were primary beneficiaries while their family members and other household decision-makers were secondary targets. Household decision-makers were often in charge of women's decisions to seek care and community leaders provided advice and financial assistance for healthcare seeking at times. Community leaders included elected representatives and teachers.

The CE activity schedule was prepared at the beginning of each month by the local study team and shared to the respective community stakeholders. Meetings lasted between 30 and 150 min. These meetings were facilitated by the study research officers with support from the medical officers at each PHC. Sessions began with a short formal introduction and description of the purpose, then discussion of CE topics prompted by informational posters developed for the project (eg, see online supplemental figure S1). Participants were encouraged to actively participate, share personal experiences and discuss case examples. There were no monetary incentives for participants to attend the meeting. Participants were provided with nutritional snacks and fruit.

### Data collection and analysis

For the purpose of assessing implementation, study staff and supervisors completed CE logs after each meeting. This included a quantitative dataset on the number of participants attending each meeting and target groups reached, as well as a qualitative dataset comprised of

community feedback. For assessing the potential impact of CE activities on health outcomes, quantitative data were collected from the prospective population-based CLIP Trial Surveillance system of pregnant women enrolled in the intervention and control groups of the study.[23 29] Data collected from the CLIP Trial Surveillance included demographic factors, birth preparedness, care seeking, obstetric information, pre-eclampsia knowledge, perinatal outcomes and treatment information. Data forms were reviewed for completeness and entered weekly into the local database. Deidentified and encrypted data were transferred to the central server for analyses.

We employed a convergent mixed-methods approach, which involves quantitative and qualitative data collection separately and integrating them at the point of interpretation to highlight the areas where the two datasets converge.[30] Figure 3 illustrates the strategy for assessing CE in the CLIP India Trial, based on the process evaluation framework for the CLIP Pilot Trial in Nigeria.[31]

► CE attendance was summarised according to participant group: pregnant women and other women of reproductive age, the pregnant women's mothers and mothers-in-law, community stakeholders (community leaders, male and female decision-makers), health workers (AWW, ANM, ASHA, ASHA supervisor, LHV, medical officers, nurses, RA) and others (family members, neighbours, friends, community members).

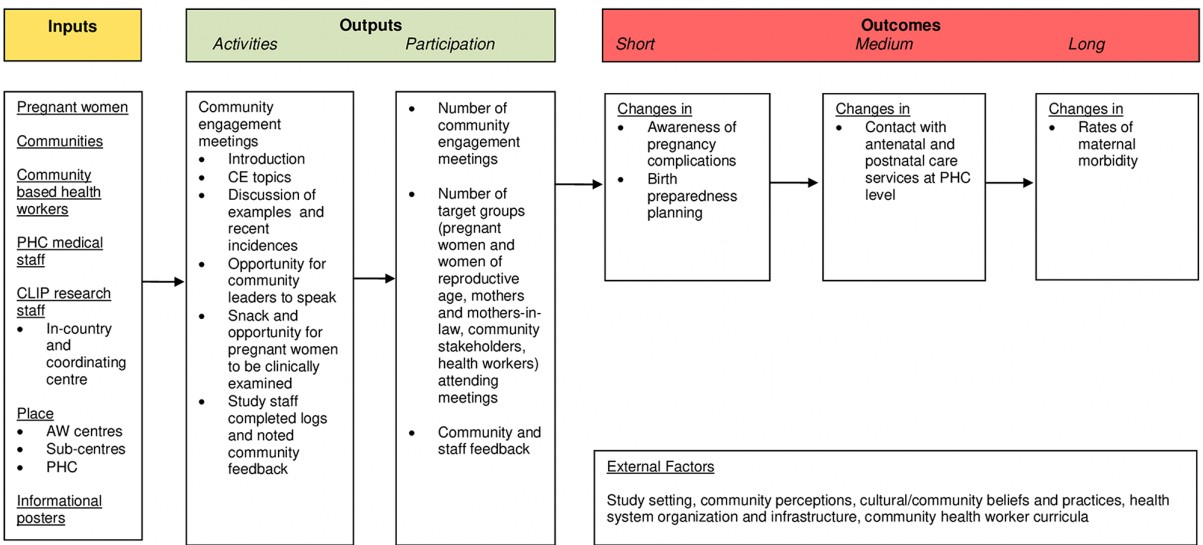

**Figure 3** A process evaluation plan for assessing CE to improve maternal health in Karnataka, India. AW, anganwadi; CE, community engagement; CLIP, Community Level Interventions for Pre-eclampsia; PHC, primary health centre.

► Pre-eclampsia knowledge was based on the clinical symptoms that predicted adverse maternal and perinatal outcomes in women with pre-eclampsia and included evaluating awareness of: (1) high blood pressure during pregnancy, (2) abnormal bleeding in pregnancy, (3) seizures in pregnancy and (4) that pregnancy hypertension can be life threatening.[32] The composite pre-eclampsia knowledge score included recalling at least one of the above conditions and at least four symptoms of high blood pressure in pregnancy.

► Birth preparedness evaluated a pregnant woman's plans for: (1) transport in case of emergencies, (2) permission to seek emergency care, (3) money saved for an emergency, (4) identified health facility for delivery and (5) at least two of the components for the BPCR composite score.

Pre-eclampsia knowledge, birth preparedness, health services engagement and perinatal outcomes indicators were summarised between arms with counts and frequencies for categorical variables and medians and IQR for continuous variables. For health service questions, we restricted the denominators to women who delivered by trial end as these indicators are more likely to occur later in pregnancy. As pre-eclampsia knowledge and birth preparedness were asked throughout pregnancy and all women received CE, we included all women in assessing these rates. Outcomes were further compared between trial arms using a mixed-effects logistic regression model (to account for clustering) with a logit link. Models were adjusted for age, maternal education, husband education, gestational age at booking and parity and results are presented as ORs and 95% CIs. Quantitative data were analysed using RStudio (RStudio, Boston, USA).

Community feedback recorded by study staff on CE logs were qualitatively analysed using a descriptive, exploratory approach to understand how communities received CE activities and supported interpretation of quantitative findings (online supplemental table S2, checklist). Feedback was recorded as open-text and included some individual comments and some group feedback given during the CE sessions. Community feedback notes were imported to NVivo V.12 software (QSR International, Melbourne, Australia) for thematic analysis.[33] Coding was conducted by qualitative experts (M-LWK, MV) in close collaboration with a community maternal and child health clinician-scientist from India). Triangulation of qualitative and quantitative data enhanced trustworthiness of findings.

### Research ethics approval
The project received ethics committee approvals from KLE Academy of Higher Education and Research Deemed-to-be-University (MDC/IECHSR/2011-12/A-4; ICMR 5/7/859/12-RHN) and the University of British Columbia (UBC, H12-03497).

### Patient and public involvement
Neither patients nor public were involved in the design or conduct of the study.

### RESULTS
### CE attendance
A total of 1379 CE meetings were conducted in the six intervention clusters. Among them 586 sessions were conducted in Belagavi district and 773 sessions at Bagalkote district. The median number of sessions per cluster was 204 (IQR: 186–281). Table 1 describes the activities across clusters and the participants reached. As long as pregnant women remained in the same village, they were encouraged to attend all sessions. The other participants, such as stakeholders and decision-makers, were usually different for each session. Pregnant women

**Table 1** Participants reached in community engagement activities in the intervention clusters

| Target groups | Number of participants (% of total) | Number of sessions (% of total) |
|---|---|---|
| Pregnant women and other women of reproductive age | 22 149 (56.3%) | 1379 (100.0%) |
| Mothers and mothers-in-law | 4643 (11.8%) | 1140 (82.7%) |
| Community stakeholders | 5685 (14.4%) | 1147 (83.2%) |
| Health workers | 4752 (12.1%) | 1376 (99.8%) |
| Others | 2113 (5.4%) | N/A |
| Total | 39 342 | 1379 |

were most represented at the meetings, followed by mothers-in-law who often accompanied them. Husbands and fathers-in-law were least represented in the meetings as they often had competing priorities; however, overall sessions were well attended by target groups.

## Impact on clinical outcomes

Tables 2 and 3 present summaries of pre-eclampsia knowledge and birth preparedness among pregnant women in the CLIP Trial. The rates for both (composite and components) were not significantly different between arms. Only a minority of participants from both intervention and control (2%–6%) could name at least four symptoms of high blood pressure in pregnancy or had overall pre-eclampsia knowledge. In contrast, there were higher levels of birth preparedness among women in both intervention and control with a majority of women reporting that they had arranged transport, has permission for emergency care and identified a health facility for delivery. Only saving funds for an obstetric emergency was lower, with about half of women declaring that they had funds saved up. Confidence intervals of odd ratios are wide, reflecting high heterogeneity between study clusters (see online supplemental tables S3 and S4).

Table 4 illustrates that health services engagement by pregnant women in the CLIP Trial were similar between arms. Almost all women had at least one ANC visit and three out of four women had at least four visits. Approximately 1 in 10 women visited the PHC for reason other than routine care and about 3% of women were admitted to a health facility for reasons other than delivery, staying for a median of 3 days. One in 20 women experienced a maternal morbidity and there was no significant difference between arms (OR 0.95, 95% CI: 0.61 to 1.50, p=0.84).

## Community feedback on CE activities
### Engaging family members

Staff noted that communities appreciated engaging household decision-makers and community stakeholders alongside pregnant women and health workers. Household decision-makers may not have been aware of pre-eclampsia and other maternal health risks. Their inclusion in the discussion supported care for pregnant women and increased value in attending ANC.

**Table 2** Summary of pre-eclampsia knowledge by arm

| | Intervention (n=7839) | Control (n=6944) | Total* (n=14 783) | Adjusted odds ratio (95% CI) | P value† |
|---|---|---|---|---|---|
| Women can have abnormal bleeding in pregnancy | 1981 (25.3%) | 2368 (34.1%) | 4349 (29.4%) | 0.88 (0.25 to 3.18) | 0.85 |
| Women can have seizure in pregnancy | 1814 (23.1%) | 1691 (24.4%) | 3505 (23.7%) | 1.49 (0.40 to 5.62) | 0.55 |
| Women can have high blood pressure in pregnancy | 3143 (40.1%) | 3073 (44.3%) | 6216 (42.1%) | 1.02 (0.32 to 3.25) | 0.97 |
| High blood pressure in pregnancy can be life threatening | 3086 (39.4%) | 2840 (40.9%) | 5926 (40.1%) | 0.86 (0.17 to 4.21) | 0.85 |
| Can name at least four symptoms | 441 (5.6%) | 153 (2.2%) | 594 (4.0%) | 2.35 (0.60 to 9.12) | 0.22 |
| Composite‡ | 439 (5.6%) | 151 (2.2%) | 590 (4.0%) | 2.35 (0.61 to 9.10) | 0.22 |

*All pregnancies enrolled in the CLIP India Trial.
†Estimated by mixed-effects logistic regression adjusted for: maternal age, maternal education, husband education, gestational age at enrolment, parity and cluster (random effect).
‡Recall at least one condition and at least four symptoms of high blood pressure.
CLIP, Community Level Interventions for Pre-eclampsia.

**Table 3** Summary of birth preparedness by arm

| | Intervention (n=7839) | Control (n=6944) | Total (n=14 783) | Adjusted odds ratio (95% CI) | P value* |
|---|---|---|---|---|---|
| Arranged transport | 6109 (77.9%) | 6045 (87.1%) | 12 154 (82.2%) | 0.40 (0.03 to 6.07) | 0.51 |
| Has permission for emergency care | 7595 (96.9%) | 6825 (98.3%) | 14 420 (97.5%) | 0.48 (0.05 to 4.31) | 0.51 |
| Has money saved for emergency | 3869 (49.4%) | 4040 (58.2%) | 7909 (53.5%) | 0.46 (0.05 to 4.19) | 0.49 |
| Identified facility for delivery | 6856 (87.5%) | 5995 (86.3%) | 12 851 (86.9%) | 1.50 (0.36 to 6.27) | 0.58 |
| Composite† | 5587 (71.3%) | 5869 (84.5%) | 11 456 (77.5%) | 0.50 (0.05 to 5.59) | 0.58 |

*All pregnancies enrolled in the CLIP India Trial.
†Estimated by mixed effects logistic regression adjusted for: maternal age, maternal education, husband education, gestational age at enrolment, parity, and cluster (random effect).
‡Having at least two of the components of birth preparedness ready.[5 7]
CLIP, Community Level Interventions for Pre-eclampsia.

The people appreciated these community engagement meetings. They said that they learnt many things about the care of the pregnant women, which they were not aware of earlier. The pregnant women became more aware about their health and their parents also started giving more support and care to her.

Earlier, it was difficult for the woman to take permission from the decision makers in the family to seek antenatal care. But after conducting the community engagement meetings, the views of the decision-makers were changed so that the seeking of permission for antenatal care became easy.

### Mobilising community support for maternal health

CE feedback indicated increased community support for BPCR. There were examples of community leaders advocating for appropriate maternal healthcare seeking with household decision-makers as well as pushing for change to support local health systems.

The views of stakeholders also changed so much that the arrangement of transport, financial support, counselling the decision-makers became effective. For example, in the initial period of the study, there was one incident where the husband, the male decision-maker of the family, of the woman with eclampsia refused to take her to the hospital but after counselling by the community leaders

and his family doctor, he agreed to take her to the higher care hospital.

The community leaders recognized the significant role played by the ASHAs and the PHC personnel in promoting the health of the pregnant women. The PHC did not have an ambulance of its own and the community leaders saw to it that a new ambulance was made available at the PHC. They also built a room above the PHC building as a meeting place for the AHSAs to continue the work actively.

### External factors and barriers

Barriers of poverty, health infrastructure gaps and poor quality of care at facilities were frequently highlighted in CE discussions.

People accepted all the topics. They only found the issue of arranging money to be a problem because they had meagre earnings.

Additionally, the cost of treatment at private hospitals was often brought up in discussions in conjunction with the lack of services at government facilities. Some participants pointed out that there was no obstetric specialist at a nearby secondary level hospital for a period of time. Staff reflected that community members sometimes highlighted the need to strengthen care at facilities in addition to raising community awareness.

**Table 4** Health services engagement and maternal morbidity by arm

| | Intervention (n=6908) | Control (n=6109) | Total* (n=13 017) | Adjusted odds ratio (95% CI) | P value† |
|---|---|---|---|---|---|
| Women with at least one ANC visit | 6891 (99.8%) | 6068 (99.3%) | 12 959 (99.6%) | Cannot be computed | – |
| Women with four or more ANC visits | 5140 (74.4%) | 4745 (77.7%) | 9885 (75.9%) | 0.86 (0.51 to 1.44) | 0.57 |
| Visited PHC for other than routine care | 823 (11.9%) | 739 (12.1%) | 1562 (12.0%) | 1.92 (0.43 to 8.55) | 0.39 |
| Visited higher level facility for other than routine care | 1069 (15.5%) | 780 (12.8%) | 1849 (14.2%) | 1.59 (0.58 to 4.38) | 0.37 |
| Number of women admitted to health facility for reasons other than delivery | 225 (3.3%) | 170 (2.8%) | 395 (3.0%) | 1.28 (0.08 to 19.3) | 0.86 |
| Median days of admission (IQR) | 3.0 (2.0 to 5.0) | 3.0 (2.0 to 5.0) | 3.0 (2.0 to 5.0) | – | – |
| Maternal morbidity‡ | 371/7839 (4.7%) | 325/6944 (4.7%) | 696/14 783 (4.7%) | 0.95 (0.61 to 1.50); | 0.84 |

*Pregnancies enrolled in the CLIP India Trial with postpartum follow-up.
†Estimated by mixed-effects logistic regression adjusted for: maternal age, maternal education, husband education, gestational age at enrolment, parity, and cluster (random effect).
‡Data taken from CLIP India primary trial paper and includes different adjustment factors.[23]
ANC, antenatal care; CLIP, Community Level Interventions for Pre-eclampsia; PHC, primary health centre.

There were some suggestions from the community about the improvement of services at the local government health centres.

At referral facilities, participants reported that they often had to wait a long time before they would be provided services. They also reported that rural, uneducated people may encounter stigma and discrimination when seeking care at urban tertiary facilities.

The participants wanted to know how to get immediate admission and care at higher and bigger tertiary care hospitals.

They also demanded about early admission and prompt service at the tertiary care hospitals as there are some uneducated persons in the community who face some problems in getting the services in bigger hospitals."

## DISCUSSION
### Summary of findings and comparisons with the literature
Our evaluation of the CLIP CE programme in India demonstrates the feasibility of reaching pregnant women alongside household decision-makers, community stakeholders and health workers. This included 1379 CE sessions reaching out to nearly 40000 stakeholders in the community. Engaging household decision-makers and community leaders together with pregnant women to discuss pregnancy care may have helped to shift local opinions on the value of ANC to monitor for pregnancy complications and mitigate risks. However, rates of pre-eclampsia knowledge, birth preparedness, health services engagement and maternal morbidities remained similar across arms in the CLIP Trial. According to our evaluation framework (figure 3), our study found no evidence that the outputs were associated with our expected outcomes.

Although research findings related to the effect of community participation to improve access to maternal healthcare services have been mixed,[34] a previous study in Uttar Pradesh in Northern India found significant effects of community health worker home visits and monthly community meetings on improved self-recognition of problems faced during pregnancy, birth preparedness components and knowledge of pregnancy danger signs.[35] Differing outcomes between our findings and the study in Uttar Pradesh highlight two issues: first, the importance of context, and second, challenges in measuring impacts of community mobilisation. Uttar Pradesh has much higher maternal mortality rates than Karnataka,[36] suggesting that CE may have a stronger effect where there is lower baseline access to maternal health services. Second, in contrast to targeted community meetings with traditional maternal-care providers and birth attendants, our CE strategy was focused on broader community mobilisation and strengthening support for maternal care seeking among community stakeholders, which has

previously been discussed in the literature as challenging to measure.[8 37]

### CE in context and potential implications for policy and practice
CE involved shifting opinions on maternal health and cultures of care, behavioural change and mobilisation and can take more time than the limited timeframe within a clinical trial. Logic models can help to understand the pathways of change, however, some of our expected short-term effects in birth preparedness via community mobilisation may take more time than measured during the trial. A systematic review found increased knowledge of BPCR as a result of interventions but increased knowledge did not always correspond to increased utilisation of maternal health services.[38] While the CE programme was successful in reaching a wide audience of participants and community feedback suggested increased support for maternal healthcare seeking, these may be upstream indicators or insufficient to change clinical outcomes.

A systematic review on the effects of community participation on improving uptake of skilled maternal and newborn care highlighted the importance of qualitative research to understand unforeseen impacts and sociopolitical factors.[34] While quantitative results did not reveal an effect in our study, staff qualitatively noted that effects were noticeable in the opinions of community leaders and household decision-makers in their support for pregnancy care. Additionally, the discrepancy between the quantitative and qualitative results may also highlight the influence of external factors where community members gained appreciation of maternal health issues but were limited in their capacity to access care. Poverty and quality of care available at local health facilities emerged as significant barriers shared in community feedback, which is in line with previous research on the determinants of maternal healthcare services utilisation in the area.[10]

Nevertheless, qualitative findings that suggested improved awareness and attitudes towards pregnancy hypertension and its complications among community leaders and household decision-makers is promising. Previous research in the area found lack of knowledge around pre-eclampsia and eclampsia.[39] Hypertension specifically as a pregnancy condition was unknown and there was no specific terminology in the local language (Kannada) for pre-eclampsia. CE has significant repercussions for the implementation of healthcare interventions and policy making.

### Strengths and limitations
Strengths of our CE programme and assessment include its systematic evaluation using a logic model framework and involvement of a wide range of health workers, pregnant women, household decision-makers and community leaders. Our study was also able to triangulate qualitative and quantitative datasets in a convergent mixed-methods approach.

However, as a secondary analysis, our evaluation of the impact of CE activities was limited. The effects of CE on community leaders and household decision-makers were not adequately captured in our evaluation framework that focused on measuring effects with only the pregnant women enrolled in the trial. Though qualitative findings revealed various themes and quotes were collected from CE sessions, they reflected comments from the group and unfortunately, we do not have individual-level CE feedback data. Additionally, the possibility of contamination cannot be ruled out as the clusters belonged to the same district. The health workers from the intervention and control clusters met at district-level meetings and may have discussed the intervention. This is a general limitation for cluster randomised settings. Furthermore, since CE was one of the components of a complex intervention, the effects assessed do not reflect CE alone. Intervention of multiple factors makes the impact assessment much more complicated.

## CONCLUSION

Our study found that CE activities may have had an effect on modifying community attitudes towards hypertension in pregnancy and its complications but did not find an impact on individual women's pre-eclampsia knowledge, birth preparedness and care seeking indicators. Contemplation of specific individual factors and community capabilities in engaging in the health seeking behaviours would contribute to the vision of supporting people-centred methodologies to deliver health promotion. Through the complex social interactions inherent in the current CE activities, appraisals over time will evaluate the effectiveness and sustainability of these interventions.

**Author affiliations**
[1]Women's and Children's Health Research Unit, J N Medical College, KLE Academy of Higher Education and Research, Belagavi, Karnataka, India
[2]Department of Obstetrics and Gynaecology, The University of British Columbia Faculty of Medicine, Vancouver, British Columbia, Canada
[3]Women's and Children's Health Research Unit, S Nijalingappa Medical College, Bagalkote, Karnataka, India
[4]Obstetrics and Gynecology, The University of British Columbia, Vancouver, British Columbia, Canada
[5]King's College London School of Medical Education, London, UK
[6]Thomas Jefferson University, Philadelphia, Pennsylvania, USA
[7]Department of Women and Children's Health, King's College London Faculty of Life Sciences and Medicine, London, UK

**Acknowledgements** We would like to thank the women and their families who contributed data to this trial, in addition to the ASHA and ANMs who deliver care to the most vulnerable populations. Also, we would like to acknowledge the key contributions of the following DSMB members: Romano Nkumbwa Byaruhanga, Brian Darlow, Eileen Hutton (Chair), and Lehana Thabane. We would also like to acknowledge the CLIP India Working Group members: Shashidhar G Bannale, Keval S Chougala, Vaibhav B Dhamanekar, Anjali M Joshi, Namdev A Kamble, Gudadayya S Kengapur, Uday S Kudachi, Sphoorthi S Mastiholi, Geetanjali I Mungarwadi, Esperança Sevene, Khátia Munguambe, Charfudin Sacoor, Eusébio Macete, Helena Boene, Felizarda Amose, Orvalho Augusto, Cassimo Bique, Ana Ilda Biz, Rogério Chiaú, Silvestre Cutana, Paulo Filimone, Emília Gonçálves, Marta Macamo, Salésio Macuacua, Sónia Maculuve, Ernesto Mandlate, Analisa Matavele, Sibone Mocumbi, Dulce Mulungo, Zefanias Nhamirre, Ariel Nhancolo, Cláudio Nkumbula, Vivalde Nobela, Rosa Pires, Corsino Tchavana, Anifa Vala, Faustino Vilanculo, Rahat N Qureshi, Sana Sheikh, Zahra Hoodbhoy, Imran Ahmed, Amjad Hussain, Javed Memon, Farrukh Raza, Olalekan O Adetoro, John O Sotunsa, Sharla K Drebit, Chirag Kariya, Mansun Lui, Diane Sawchuck, Ugochi V Ukah, Mai-Lei Woo Kinshella, Shafik Dharamsi, Guy A Dumont, Tabassum Firoz, Ana Pilar Betrán, Susheela M Engelbrecht, Veronique Filippi, William A Grobman, Marian Knight, Ana Langer, Simon A Lewin, Gwyneth Lewis, Craig Mitton, Nadine Schuurman, James G Thornton, France Donnay, Romano N Byaruhanga, Brian Darlow, Eileen Hutton, Mario Merialdi, Lehana Thabane and Kelly Pickerill.

**Contributors** PvD, RD, MB, SSG, AM and LM conceptualised the trial and components of the intervention. MB, SSG, AM, UC, GMK, UYR, AK, NVH, CCK, SGB, MV and GIM implemented and coordinated monitoring of the trial. JNB and M-LWK performed the quantitative and qualitative analyses. AK and M-LWK wrote the first draft of the manuscript. PvD, MB, SSG are the overall as guarantors for the data. All authors provided feedback and review of the manuscript.

**Funding** This trial was funded by the University of British Columbia, a grantee of the Bill & Melinda Gates Foundation (PRE-EMPT initiative, grant number OPP1017337). Following input into trial design, the Gates Foundation had no role in data collection, analysis, or interpretation or writing of the report.

**Map disclaimer** The inclusion of any map (including the depiction of any boundaries therein), or of any geographic or locational reference, does not imply the expression of any opinion whatsoever on the part of BMJ concerning the legal status of any country, territory, jurisdiction or area or of its authorities. Any such expression remains solely that of the relevant source and is not endorsed by BMJ. Maps are provided without any warranty of any kind, either express or implied.

**Competing interests** None declared.

**Patient and public involvement** Patients and/or the public were not involved in the design, or conduct, or reporting or dissemination plans of this research.

**Patient consent for publication** Not applicable.

**Ethics approval** The project received ethics committee approvals from KLE Academy of Higher Education and Research Deemed-to-be-University (MDC/IECHSR/2011-12/A-4; ICMR 5/7/859/12-RHN) and the University of British Columbia (UBC, H12-03497).

**Provenance and peer review** Not commissioned; externally peer reviewed.

**Data availability statement** Data are available upon reasonable request. Not Applicable.

**ORCID iDs**
Avinash Kavi http://orcid.org/0000-0002-2176-4697
Mai-Lei Woo Kinshella http://orcid.org/0000-0001-5846-3014
Umesh Charantimath http://orcid.org/0000-0003-1726-2798
Jeffrey N Bone http://orcid.org/0000-0001-7704-1677
Marianne Vidler http://orcid.org/0000-0002-7633-8812
Shivaprasad S Goudar http://orcid.org/0000-0002-8680-7053
Mrutyunjaya Bellad http://orcid.org/0000-0003-0460-1439
Peter von Dadelszen http://orcid.org/0000-0003-4136-3070

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
