## [Reviewer comments · BMJ Open]

ARTICLE DETAILS

TITLE (PROVISIONAL)	Community engagement for birth preparedness and complication readiness in the Community Level Interventions for Pre-eclampsia (CLIP) trial in India: A mixed method evaluation
AUTHORS	Kavi, Avinash; Kinshella, Mai-Lei Woo; Ramadurg, Umesh Y.; Charantimath, Umesh; Katageri, Geetanjali M.; Karadiguddi, Chandrashekhar C.; Honnunar, Narayan V.; Bannale, Shashidhar G.; Mungarwadi, Geetanjali I.; Bone, Jeffrey; Vidler, Marianne; Magee, Laura; Mallapur, Ashalata; Goudar, Shivaprasad S.; Bellad, Mrutyunjaya; Derman, Richard; von Dadelszen, Peter; The CLIP India Working Group, .

VERSION 1 – REVIEW

REVIEWER	Ahmad, Danish University of Canberra Faculty of Health, Health Research institute
REVIEW RETURNED	16-Mar-2022

GENERAL COMMENTS	Thank you for the opportunity to review the paper. The authors have conducted a mixed-method study to identify the effect of a community intervention programme on participating women's open complication readiness for preeclampsia in a rural state of India. Data from 1400 community meetings involving approximately 40,000 participants was analysed to identify changes in short term outcomes related to knowledge of a clumsier both preparedness and health system engagement. A logic framework was used as the evaluation framework for the study. The authors identify a lack of significant shift in outcomes due to programme activities. The authors and the programme teams seek to improve maternal health outcomes in rural India by adapting a community health programme that has been used globally elsewhere and has been published. The paper in its current form requires revisions in the study context methodology and presentation of results to be accepted for publication. Comment 1. Background: The authors are on the right track to identify delays in seeking care as a reason for high maternal mortality globally? The introduction section does not adequately address or identify the research gap India has progressed in maternal health
--

	indicators by implementing national-level programmes that have improved institution deliveries as well as the provision of frontline workers to provide maternal healthcare. Could the authors provide details specific to the region where the studies were conducted in Karnataka in terms of the burden of disease in this state relative to the country and also provided a short summary Of how this programme integrates with existing government health services? 2. In the methods section, some details of the CLIP intervention are provided; however, it was not evident clearly what the programme essentially was who implemented it end what activity was conducted in these meetings. For example, did the meetings develop health knowledge about preeclampsia by using IEC material Or some other methodology? Additionally, some details of sampling of intervention and control areas are required here, even though they may have been published elsewhere. 3. How did the study establish equivalence between the intervention and the comparison areas. 4. While the study employs mixed methods, the method section does not adequately explain the epidemiology of the qualitative study and how the qualitative and quantitative study objectives are related. 5. The authors may like to add a subsection specific to the quantitative study under methods. 6. I found the results to be insightful, but the presentation of results did not integrate well between the qualitative and quantitative components. Again coming to the discussion section, it wasn't easy to identify at what specific level that programme had been implemented in who the primary and secondary intended targets were. it can be assumed that women of reproductive age groups were primary beneficiaries while their family members another household decision-makers were secondary targets. 7. The authors would like to refer to the World Health Organisation definition of BPCR and use it in their paper. 8. Can the authors please provide some rationale references to the construction of the preeclampsia knowledge variable? I wasn't sure how awareness of having abnormal bleeding in pregnancy was linked to preeclampsia which tends to manifest largely with hypertensive symptoms for women
--	--

REVIEWER	Doddihal, Chandrika BLDE University, Community Medicine
REVIEW RETURNED	05-Apr-2022

GENERAL COMMENTS	The article is well written with adequate mention of research question, objectives and outcomes. The abstract however, can include the research methodology more clearly (inclusion of randomization and controls). The probable reason for non significant results could be contamination between intervention and control clusters which can be discussed.
---

REVIEWER	Challen, Robert University of Exeter, EPSRC Centre for Predictive Modelling in Healthcare
REVIEW RETURNED	08-Jul-2022

GENERAL COMMENTS	Evaluating the effect of community engagement on the birth preparedness and complication readiness in the Community Level Interventions for Pre-eclampsia (CLIP) randomised trial in northern Karnataka, India This is an interesting paper describing the findings of a cluster randomised control study into maternal outcomes following a community education programme intervention. It is well written but at 116 pages it is pretty daunting for a reviewer and I feel a closer adherence to CONSORT guidelines would have made my job simpler. Statistical review: Design: CRT Clusters: - geographical (PHC level) - 6 clusters control; 6 clusters intervention Intervention: Community engagement; blood pressure assessment by Accredited Social Health Activists (ASHA); and usage of PIERS-On-the-Move (POM) mobile-based decision aid. Outcome: Pre-eclampsia knowledge, birth preparedness, health services engagement and perinatal outcomes indicators. Confounders: age, maternal education, husband education, gestational age at booking, and parity Method: mixed effects logistic regression model Comments: Background: What is the existing standard of care for pregnancy in India, to what extent does the community engagement and monitoring represent a change? Major: Given the size of the supplementary you cannot assume people will have read it in detail. If there is any important information in it it needs to be brought into the main paper. Much of the supplementary protocol is irrelevant to the study at hand and seems to be about the wider CLIP study. A much shorter summary of the protocol specifically as it applies to this study in India would have been greatly appreciated. Methods: P7 line 10: more detail on the size of population of each cluster for those not familiar with the CLIP trial. What is a PHC area - how many villages? Page 7 line 12: Is it 12 clusters control; 12 clusters intervention or 6 clusters control and 6 clusters intervention. Looks like the latter based on a trawl through the protocol but probably ought to be clearer in the main paper. Figure 1 does not describe the smaller subdivisions are these the PHC areas?. Data collection: The data collection seems to be part of the wider CLIP trial. Do we know anything about the completeness of enrollment? Is there any risk of incomplete enrollment varying by cluster? Were all pregnancies identified in every cluster? Major: I need to see more information about the clusters stratified by their randomisation. I want to know for example are the intervention and control clusters evenly geographically distributed? Is the population in those clusters very different? What is the average age, socioeconomic deprivation, urban / rural mix, provision of healthcare facilities. It would be good to see the location of the clusters on a map.
---

Is there risk of differential response rates to CLIP questionnaires in intervention and control group? Could the community engagement have improved response rate to questionnaire?

Results:

Major: You don't seem to describe how many women are in both intervention and control arms (until table 4). You discuss the number of engagements and the number of women that attended the engagements, but not the number of women who responded to the CLIP questionnaires in intervention and control groups. Are the people who attended the sessions the same as the people who responded to the CLIP questionnaires? I expect the results to start with the denominator numbers from table 4, not the number of attendances at the community sessions.

A CONSORT flow chart detailing the patient recruitment, randomisation, any exclusions, and loss to follow up is highly desirable, maybe as supplementary.

See section 13 here for example: <https://www.bmj.com/content/340/bmj.c869>

Major: Table 1

The CONSORT demographic table is missing here, your table 1 seems to be something else. The demographic table should detail the number of participants in each arm and describe and compare the distribution of the confounders in both arms (ideally with a statistical test). Particularly age, maternal education, husband education, gestational age at booking, and parity are mentioned. It should look like this:

		Intervention N = ??	Control N = ??	Sign value
Age years, (median + IQR)		?? [?? - ??]	?? [?? - ??]	<0.0
Maternal education level (Count)	High	??	??	<0.0
	Medium	??	??	
	Low	??	??	
Gestation age at booking (median + IQR)		?? [?? - ??]	?? [?? - ??]	<0.0

See section 15 and the guidance here: <https://www.bmj.com/content/340/bmj.c869>

Major: Tables 2 and 3

You don't adequately explain the denominator of the population responding to questionnaires in the table 2 and 3. I think the denominators in e.g. the first line of table 2 are $1981/0.253 = 7830$ for intervention, $2368/0.341 = 6944$ for control and $4349/0.294 = 14792$ total. Oddly the 7830 and 6944 don't add up to 14792. Please could you report the denominator for each group. There is an example of this in Section 17a and 17b in <https://www.bmj.com/content/340/bmj.c869>

If your denominators are not the same for the whole table because of missing data please can you highlight reasons for missing data and comment on the impact.

You do report the denominators for table 4 which is much clearer. Could you make table 2 &

3 look like table 4?

Minor: Does the reporting format for odds ratios and p-values conform to the journal's standard? I'd normally put p-value in a separate column.

Minor: Could you consider reporting the full logistic regression models of the key elements of tables 2-4 in the supplementary.

In summary:

- Please clarify what the existing standard of care is for the control groups.
- Please report some demographic information about the control and intervention clusters to reassure me they are approximately comparable.
- Please report on numbers at each stage of the trial, ideally with a flowchart.

	 ● Please redo table 1 to fit with CONSORT guidance. ● Please report denominators in table 2 & 3
--	--

VERSION 1 – AUTHOR RESPONSE

Reviewer 1:

Thank you for the opportunity to review the paper.

The authors have conducted a mixed-method study to identify the effect of a community intervention programme on participating women's open complication readiness for preeclampsia in a rural state of India. Data from 1400 community meetings involving approximately 40,000 participants was analysed to identify changes in short term outcomes related to knowledge of a clumsier both preparedness and health system engagement. A logic framework was used as the evaluation framework for the study. The authors identify a lack of significant shift in outcomes due to programme activities.

The authors and the programme teams seek to improve maternal health outcomes in rural India by adapting a community health programme that has been used globally elsewhere and has been published. The paper in its current form requires revisions in the study context methodology and presentation of results to be accepted for publication.

Reply: Thank you for the insightful review of the paper and providing the highlight of the paper.

Comments:

✓ 1. Background: The authors are on the right track to identify delays in seeking care as a reason for high maternal mortality globally. The introduction section does not adequately address or identify the research gap India has progressed in maternal health indicators by implementing national-level programmes that have improved institution deliveries as well as the provision of frontline workers to provide maternal healthcare. Could the authors provide details specific to the region where the studies were conducted in Karnataka in terms of the burden of disease in this state relative to the country and also provided a short summary of how this programme integrates with existing government health services?

Reply: Thank you for pointing out this important issue, which adds to the value of the paper. The background section has been thoroughly edited to include a section on the burden of maternal mortality in Karnataka state relative to the country and a section on how community engagement supports the existing government frontline health worker programme. Bringing community members together to raise awareness of maternal health issues can help mobilize wider support for maternal healthcare seeking and the work of community health workers.

✓ 2. In the methods section, some details of the CLIP intervention are provided; however, it was not evident clearly what the programme essentially was who implemented it end what activity was conducted in these meetings. For example, did the meetings develop health knowledge about preeclampsia by using IEC material or some other methodology? Additionally, some details of sampling of intervention and control areas are required here, even though they may have been published elsewhere.

Reply: As rightly mentioned, we have elaborated on the details of the community engagement activities. First, we have added more detail on the CE program, specifically about the informational posters used to prompt discussion. These are now referenced in the manuscript and an example added to the supplementary files (Figure S1). Secondly, we have also added details of the cluster selection and randomization procedures. The primary health centre and its catchment area served as the unit of randomization. Twelve clusters were chosen according to region (six in each district),

accessibility for surveillance, and absence of conflicting research activity. Clusters were allocated using a restricted randomisation algorithm, balancing region and population size to ensure equivalence between intervention and control clusters. Stratified randomisation was undertaken to allocate six clusters each to the intervention and control groups, using population size as a single stratification factor.

✓ 3. How did the study establish equivalence between the intervention and the comparison areas.

Reply: Thank you for raising this important issue. We have added more description of the randomisation process, as detailed in the response to the above comment. Clusters were allocated by the study statistician using a restricted randomisation algorithm, balancing region and population size to ensure equivalence between intervention and control clusters.

✓ 4. While the study employs mixed methods, the method section does not adequately explain the epidemiology of the qualitative study and how the qualitative and quantitative study objectives are related. The authors may like to add a subsection specific to the quantitative study under methods.

Reply: Thank you for highlighting this gap in our methodology description. We have reworked the description to better clarify the sources of qualitative and quantitative data. From the CLIP Trial surveillance system, we collected quantitative data on demographic factors, birth preparedness, care-seeking, obstetric information, pre-eclampsia knowledge, perinatal outcomes and treatment information. After each CE session, study staff and supervisors documented the number of participants and target groups for quantitative analysis and community feedback for qualitative analysis. We added that our study employed a convergent mixed-methods approach, which involves quantitative and qualitative data collection separately and integrating them at the point of interpretation to highlight the areas where the two datasets converge. Qualitative and quantitative data are both used for the paper's objective to describe the process of community engagement in northern Karnataka, India and its impact on pre-eclampsia knowledge, birth preparedness and complication readiness, pregnancy-related care seeking and maternal morbidity. The qualitative data shed light on how communities received the CE activities, which helped interpretation of the quantitative data.

✓ 5. I found the results to be insightful, but the presentation of results did not integrate well between the qualitative and quantitative components.

Reply: Thank you for highlighting the insightfulness of the results and the opportunity to clarify. We have reworked the presentation of the results section with the quantitative findings reported first, then a sub-section on "Community feedback on community engagement activities" for the qualitative findings. The qualitative findings have been reorganized as three themes for clarity.

✓ 6. Again coming to the discussion section, it wasn't easy to identify at what specific level that programme had been implemented in who the primary and secondary intended targets were. It can be assumed that women of reproductive age groups were primary beneficiaries while their family members another household decision-makers were secondary targets.

Reply: Thank you for highlighting the insight from the results. As rightly pointed out that pregnant women and women of reproductive age groups were primary beneficiaries while their family members another household decision-makers were secondary targets. This has been made clear in the methods section.

✓ 7. The authors would like to refer to the World Health Organisation definition of BPCR and use it in their paper.

Reply: Thank you for the recommendation. This has now been added.

✓ 8. Can the authors please provide some rationale references to the construction of the preeclampsia knowledge variable? I wasn't sure how awareness of having abnormal bleeding in pregnancy was linked to preeclampsia which tends to manifest largely with hypertensive symptoms for women

Reply: We have provided a reference as well as elaborated that the pre-eclampsia knowledge variable was based on the clinical symptoms that predicted adverse maternal and perinatal outcomes

in women with pre-eclampsia. The presence of vaginal bleeding with abdominal pain was found to be significantly associated adverse maternal outcomes among women with pre-eclampsia.

- Yen TW, Payne B, Qu Z, Hutcheon JA, Lee T, Magee LA et al. Using clinical symptoms to predict adverse maternal and perinatal outcomes in women with preeclampsia: data from the PIERS (Pre-eclampsia Integrated Estimate of RiSk) study. J Obstet Gynaecol Can 2011; 33(8):803-809.

Reviewer 2:

Comments to the Author:

The article is well written with adequate mention of research question, objectives and outcomes.

Reply: Thank you for the positive feedback.

✓ 1. The abstract however, can include the research methodology more clearly (inclusion of randomization and controls).

Reply: Methods section of abstract has been updated with additional information. We have added that there were a total of 12 clusters based on primary health centre catchment areas were randomized to intervention or control. Community engagement was conducted in intervention clusters.

✓ 2. The probable reason for non-significant results could be contamination between intervention and control clusters which can be discussed.

Reply: As rightly mentioned, we also agree that there might be a chance of contamination between intervention and control clusters as they were located in a similar district. We have included that in the discussion.

Reviewer 3:

This is an interesting paper describing the findings of a cluster randomised control study into maternal outcomes following a community education programme intervention. It is well written but at 116 pages it is pretty daunting for a reviewer and I feel a closer adherence to CONSORT guidelines would have made my job simpler.

/

Statistical review:

- Design: CRT
- Clusters - geographical (PHC level) - 6 clusters control; 6 clusters intervention
- Intervention: Community engagement; blood pressure assessment by Accredited Social Health Activists (ASHA); and usage of PIERS-On-the-Move (POM) mobile-based decision aid.
- Outcome: Pre-eclampsia knowledge, birth preparedness, health services engagement and perinatal outcomes indicators.

- Confounders: age, maternal education, husband education, gestational age at booking, and parity
- Method: mixed effects logistic regression model

Reply: Thank you for a positive feedback. We have now included the CONSORT checklist for greater clarity as well as highlighted that this was a secondary analysis of the CLIP Trial in India. The protocol was requested as part of the submission to the journal but we have now developed a shorter version including specifically the community engagement activities, with reference to the full protocol.

Comments:

Background:

✓ 1. What is the existing standard of care for pregnancy in India, to what extent does the community engagement and monitoring represent a change?

Reply: We have added a description of the existing standard of care in India to the background section. Pregnant women are identified by community health workers called Accredited Social Health Activists (ASHAs). ASHAs accompany pregnant women to a nearby primary health centre for registration and antenatal visits. However, previous research in Karnataka revealed that pregnant women rarely made health care decisions on their own, and other family members and community leaders were highly influential in decision-making. Bringing community members together to raise awareness of maternal health issues mobilize wider support for maternal healthcare seeking and ASHAs' work in local communities.

✓ 2. Given the size of the supplementary you cannot assume people will have read it in detail. If there is any important information in it needs to be brought into the main paper.

Much of the supplementary protocol is irrelevant to the study at hand and seems to be about the wider CLIP study. A much shorter summary of the protocol specifically as it applies to this study in India would have been greatly appreciated.

Reply: Thank you for highlighting this issue and we completely agree that the full CLIP protocol is lengthy. The protocol was requested as part of the submission to the journal but we have now developed a shorter version including specifically the community engagement activities, with reference to the full protocol.

Methods:

✓ 3. P7 line 10: more detail on the size of population of each cluster for those not familiar with the CLIP trial. What is a PHC area - how many villages?

Reply: We have added more detail on the study setting, adding that this study was conducted in 12 distinct geographic regions in Karnataka State: six in Belagavi District and six in neighbouring Bagalkote District. Belagavi has a population of 4,779,661 living in 1278 villages and Bagalkote has a population of 1,889,752 living in 627 villages. There are an estimated 500 births annually in each PHC area (cluster).

✓ 4. Page 7 line 12: Is it 12 clusters control; 12 clusters intervention or 6 clusters control and 6 clusters intervention. Looks like the latter based on a trawl through the protocol but probably ought to be clearer in the main paper.

Reply: Thank you for highlighting this ambiguity. We have clarified that six clusters were randomised each to the intervention and control groups.

✓ 5. Figure 1 does not describe the smaller subdivisions are these the PHC areas?

Reply: Map depicting detailed PHC area has been added.

✓ 6. Data collection: The data collection seems to be part of the wider CLIP trial. Do we know anything about the completeness of enrolment? Is there any risk of incomplete enrolment varying by cluster? Were all pregnancies identified in every cluster?

Reply: The data was collected as part of CLIP Trial Surveillance, which consisted of maternal and newborn health registry implemented with the trial. All pregnancies identified in every cluster as part of the registry and all the consenting women were enrolled into the trial. There was no major difference in the completeness of the enrolment among the cluster as there were very few women who dropped out in both intervention and control clusters.

✓ 7. Major: I need to see more information about the clusters stratified by their randomisation. I want to know for example are the intervention and control clusters evenly geographically distributed? Is the population in those clusters very different? What is the average age, socioeconomic deprivation, urban / rural mix, provision of healthcare facilities. It would be good to see the location of the clusters on a map.

Reply: Total 12 clusters (6 each from Belagavi and Bagalkote districts of Karnataka) were identified with approximately 500 birth annually, each covering a population of ~ 30,000. Among them 6 intervention and 6 control clusters (3 each in each district) were stratified by their randomisation. Maps depicting detailed PHC area has been added (see Figures 1 and 2). We have also added the following line, "Both intervention and control clusters were similar in terms of population (11.6 per hectare in intervention clusters vs 13.3 per hectare in control clusters) and community health workers (17 vs 18 ASHAs per cluster) density. Among the pregnant women enrolled in the CLIP Trial (7839 in intervention clusters, 6944 in the control clusters), maternal age was on average 22-23 years old, 56-58% of women had 8 or more years of schooling, and populations were over 90% Hindu without significant difference between intervention and control clusters."

✓ 8. Is there risk of differential response rates to CLIP questionnaires in intervention and control group? Could the community engagement have improved response rate to questionnaire?

Reply: While we expected a differential response rate to CLIP Trial Surveillance, there was not a significant difference between the intervention and control groups.

Results:

✓ 9. Major: You don't seem to describe how many women are in both intervention and control arms (until table 4). You discuss the number of engagements and the number of women that attended the engagements, but not the number of women who responded to the CLIP questionnaires in intervention and control groups. Are the people who attended the sessions the same as the people who responded to the CLIP questionnaires? I expect the results to start with the denominator numbers from table 4, not the number of attendances at the community sessions.

Reply: Thank you for highlighting this critical issue. First, we have highlighted that the current paper is a secondary analysis in the abstract and introduction. Secondly, we have reworked the data collection

and analysis description in the methods for greater clarity, “For the purpose of assessing implementation, study staff and supervisors completed CE logs after each meeting. This included a quantitative dataset on the number of participants attending each meeting and target groups reached, as well as a qualitative dataset comprised of community feedback. For assessing the potential impact of CE activities on health outcomes, quantitative data was collected from the CLIP Trial Surveillance including pregnant women from both intervention and control enrolled in the study (23,29). Data collected from the CLIP Trial Surveillance included demographic factors, birth preparedness, care-seeking, obstetric information, pre-eclampsia knowledge, perinatal outcomes and treatment information.” Information about how many women are in both intervention and control arms have been added to the description of the overall trial and as denominators in Tables 2 and 3 (see response to comment 10 and 11).

✓ 10. A CONSORT flow chart detailing the patient recruitment, randomisation, any exclusions, and loss to follow up is highly desirable, maybe as supplementary.

See section 13 here for example: <https://www.bmj.com/content/340/bmj.c869>

Major: Table 1

The CONSORT demographic table is missing here, your table 1 seems to be something else. The demographic table should detail the number of participants in each arm and describe and compare the distribution of the confounders in both arms (ideally with a statistical test). Particularly age, maternal education, husband education, gestational age at booking, and parity are mentioned. It should look like this:

Intervention N = ??

Control N = ??

Significance (p value)

Age years, (median + IQR) ?? [?? - ??] ?? [?? - ??] <0.05

Maternal education level (Count)

High ?? ?? <0.05

Medium ?? ??

Low ?? ??

Gestation age at booking (median+ IQR?? [?? - ??] ?? [?? - ??] <0.05

See section 15 and the guidance here: <https://www.bmj.com/content/340/bmj.c869>

Reply: Thank you for the constructive feedback. The CONSORT flow chart is published in the primary paper. However, we agree with the comment that more information is required in the current paper and have added a summary of the participants enrolled, “Among the pregnant women enrolled in the CLIP Trial (7,839 in intervention clusters, 6,944 in the control clusters), maternal age was on average 22-23 years old, 56-58% of women had 8 or more years of schooling, and populations were over 90% Hindu without significant difference between intervention and control clusters (23). The primary paper is referenced for more information.

✓ 11. Major: Tables 2 and 3

You don't adequately explain the denominator of the population responding to questionnaires in the table 2 and 3. I think the denominators in e.g. the first line of table 2 are $1981/0.253 = 7830$ for intervention, $2368/0.341 = 6944$ for control and $4349/0.294 = 14792$ total. Oddly the 7830 and 6944 don't add up to 14792. Please could you report the denominator for each group. There is an example of this in Section 17a and 17b in <https://www.bmj.com/content/340/bmj.c869>. If your denominators are not the same for the whole table

because of missing data, please can you highlight reasons for missing data and comment on the impact. You do report the denominators for table 4 which is much clearer. Could you make table 2 & 3 look like table 4?

Reply: Thank you for the opportunity to clarify. We have added the denominators for table 2 and 3 similar to table 4, as recommended. We have created a separate sub-section of results for “Impact on clinical outcomes” and correspondingly clarified in the methods that clinical outcome data are pulled from the CLIP Trial Surveillance dataset that covers all pregnancies enrolled in the study. To further clarify, we have added footnotes to table 2, 3 and 4 regarding origin of the denominators.

✓ 12. Minor: Does the reporting format for odds ratios and p-values conform to the journal’s standard? I’d normally put p-value in a separate column.

Reply: P-value is now in a separate column in Tables 2, 3 and 4.

✓ 13. Minor: Could you consider reporting the full logistic regression models of the key elements of tables 2-4 in the supplementary.

Reply: Thank you for the suggestion. We have elected to report the logistic regression models in the same way as the published primary paper for continuity. We did discuss with our team and decided to keep it aligned with the primary paper.

In summary:

- Please clarify what the existing standard of care is for the control groups.
- Please report some demographic information about the control and intervention clusters to reassure me they are approximately comparable.
- Please report on numbers at each stage of the trial, ideally with a flowchart.
- Please redo table 1 to fit with CONSORT guidance.
- Please report denominators in table 2 & 3

Reply: Thank you for the thoughtful and constructive feedback. We have detailed our responses to these major comments, as well as the minor comments above.

VERSION 2 – REVIEW

REVIEWER	Ahmad, Danish University of Canberra Faculty of Health, Health Research institute
REVIEW RETURNED	26-Sep-2022

GENERAL COMMENTS	The paper is much approved and I can appreciate the authors have worked with three reviewers to address comments. I am suggesting a minor revision based on added details required for the qualitative methods-you mentioned 'In qualitative the dataset comprised of community feedback'-I couldn't locate specific qualitative data collection methods and from whom data was obtained. Please clarify these in the methods
---

REVIEWER	Challen, Robert University of Exeter, EPSRC Centre for Predictive Modelling in Healthcare
REVIEW RETURNED	29-Sep-2022

GENERAL COMMENTS	The authors have clarified that this is a follow up analysis of a previously published study of the CLIP trial from Pregnancy Hypertension (Bellad 2020). The previous publication covers many of the issues in terms of missing detail from this paper. However this does mean that the analysis presented here is only really interpretable in the context of the Bellad 2020 article. Given that this is the case I wonder why the Bellad 2020 is reference number 23 and not reference number 1. I would suggest that the very first sentence sets the context that this is a follow up analysis of previously published study data and signposts the user to the Bellad 2020 paper, as reference number 1.
--

VERSION 2 – AUTHOR RESPONSE

Reviewer: 1

Dr. Danish Ahmad, University of Canberra Faculty of Health, Public Health Foundation of India

Comments to the Author:

The paper is much approved and I can appreciate the authors have worked with three reviewers to address comments. I am suggesting a minor revision based on added details required for the qualitative methods-you mentioned 'In qualitative the dataset comprised of community feedback'- I couldn't locate specific qualitative data collection methods and from whom data was obtained. Please clarify these in the methods

Reply: Thank you for highlighting this gap in describing the qualitative methodologies. We have added a paragraph in the methods to better describe the qualitative data collection and analysis methods. We have also reported according to the Standards for Reporting Qualitative Research (SRQR) checklist to ensure comprehensiveness of our descriptions (Table S2).

Reviewer: 3

Dr. Robert Challen, University of Exeter, Taunton and Somerset NHS Foundation Trust

Comments to the Author:

The authors have clarified that this is a follow up analysis of a previously published study of the CLIP trial from Pregnancy Hypertension (Bellad 2020). The previous publication covers many of the issues in terms of missing detail from this paper. However this does mean that the analysis presented here is only really interpretable in the context of the Bellad 2020 article.

Given that this is the case I wonder why the Bellad 2020 is reference number 23 and not reference number 1.

I would suggest that the very first sentence sets the context that this is a follow up analysis of previously published study data and signposts the user to the Bellad 2020 paper, as reference number 1.

Reply: Thank you for the suggestion. We have not changed anything for this comment as per the editor's comment.

Reviewer: 1

Competing interests of Reviewer: Nil

Reviewer: 3

Competing interests of Reviewer: None

Thank you. Please let us know if you need anything else.

Best regards,

Avinash Kavi, on behalf of the authors